# Retinal Thickness Correlates with Cerebral Hemodynamic Changes in Patients with Carotid Artery Stenosis

**DOI:** 10.3390/brainsci12080979

**Published:** 2022-07-25

**Authors:** William Robert Kwapong, Junfeng Liu, Jincheng Wan, Wendan Tao, Chen Ye, Bo Wu

**Affiliations:** Neurology Department, West China Hospital of Sichuan University, No. 37 Guo Xue Xiang, Chengdu 610041, China; big_will_kwap@hotmail.com (W.R.K.); junfengliu225@outlook.com (J.L.); 2019324025225@stu.scu.edu.cn (J.W.); taowendan@wchscu.cn (W.T.); yechen11@stu.scu.edu.cn (C.Y.)

**Keywords:** carotid artery stenosis, computed tomography perfusion, cerebral hemodynamics, retinal thickness, choroid

## Abstract

Background: We aimed to assess the retinal structural and choroidal changes in carotid artery stenosis (CAS) patients and their association with cerebral hemodynamic changes. Asymptomatic and symptomatic patients with unilateral CAS were enrolled in our study. Material and methods: Swept-source optical coherence tomography (SS-OCT) was used to image the retinal nerve fiber layer (RNFL), ganglion cell-inner plexiform layer (GCIPL), while SS-OCT angiography (SS-OCTA) was used to image and measure the choroidal vascular volume (CVV) and choroidal vascular index (CVI). Computed Tomography Perfusion (CTP) was used to assess the cerebral perfusion parameters; relative perfusion (r) was calculated as the ratio of the value on the contralateral side to that on the ipsilateral side. Results: Compared with contralateral eyes, ipsilateral eyes showed significantly thinner RNFL (*p* < 0.001), GCIPL (*p* = 0.013) and CVV (*p* = 0.001). Relative cerebral blood volume (rCBV) showed a significant correlation with RNFL (*p* < 0.001), GCIPL (*p* < 0.001) and CVI (*p* = 0.027), while the relative permeability surface (rPS) correlated with RNFL (*p* < 0.001) and GCIPL (*p* < 0.001). Conclusions: Our report suggests that retinal and choroidal changes have the potential to detect hemodynamic changes in CAS patients and could predict the risk of stroke.

## 1. Introduction

Carotid artery stenosis (CAS), a major cause of cerebral microcirculation dysfunction [1,2], is reported to cause 20% of all ischemic strokes [3,4]. A total of 10–15% of all new strokes that arise are due to untreated CAS [5]; thus, timely diagnosis and treatment may help reduce the occurrence of ischemic stroke and its complications. With the increasing prevalence of CAS, screening for abnormalities preceding the incidences of CAS may be of clinical importance.

Neuroimaging tools such as magnetic resonance imaging (MRI) and PET perfusion are shown to be useful in imaging and evaluating the cerebral abnormalities in CAS [6,7,8]; nonetheless, these neuroimaging modalities are expensive, time-consuming, and cannot be used for large-scale screening. In the clinical setting, CT perfusion (CTP) is commonly used to assess cerebral perfusion in CAS patients [9]; however, most patients are unable to tolerate it because of its limitations. Thus, there is a need for simpler, non-invasive and cheaper tools to enable effective screening of CAS.

As the ophthalmic artery is a branch of the internal carotid artery, ophthalmic manifestations such as ocular ischemic syndrome and amaurosis fugax are frequently reported in patients with CAS [10]. Several imaging tools have been utilized to assess ocular changes associated with CAS; color doppler imaging [11,12] and fluorescein angiography [13] reports showed reduced blood flow in CAS patients compared with controls. Recent reports [14,15,16] suggest the optical coherence tomography (OCT)/OCT angiography (OCTA) as a suitable modality to explore CAS pathology of the retina as an extension of the brain because it is non-invasive and has a high resolution compared to previous ophthalmic imaging modalities. To date, reports on CAS were confined to the retinal and choroidal changes, while very little is known about the association between these changes and their cerebral hemodynamic changes.

Using the swept-source OCT(SS-OCT)/SS-OCTA, our current study aimed to assess the association between retinal and choroidal changes with cerebral hemodynamic changes in CAS patients. 

## 2. Methods

Our study enrolled unilateral asymptomatic or symptomatic CAS patients from the Neurology Department of West China Hospital, Sichuan University from December 2020 to January 2022. The study was approved by the Ethics Committee of West China Hospital, Sichuan University (Approval number 2019881) and followed the Declaration of Helsinki. All patients signed and provided informed consent. Diagnosis of stenosis in the internal carotid artery was detected with head-and-neck computed tomographic angiography (CTA). In our study, the eye on the stenosed carotid artery was described as the ipsilateral eye, while the non-stenosed carotid artery was described as the contralateral eye.

The inclusion criteria for our patients were as follows: 1. Age ≥ 18 years; 2. Carotid artery stenosis ≥ 50%; 3. Individuals who could cooperate and complete CTP and SS-OCT/SS-OCTA imaging; 4. Participants with carotid artery stenosis of 50% or more confirmed on CTA. The exclusion criteria were as follows: 1. Participants with previous ocular surgery (<6 months); 2. Participants with myopia (more than 6 diopters); 3. Participants with neurodegenerative diseases such as Alzheimer’s disease, Parkinson’s disease and multiple sclerosis; 4. Presence of cerebral hemorrhage on magnetic resonance imaging 5. The cerebral perfusion status in the ipsilesional middle cerebral artery (MCA) territory being normal, i.e., comparable to the contralateral side, defined as a relative difference in cerebral blood flow (CBF) ≤ 30% between bilateral MCA territories on CT perfusion (CTP) maps; 6. Presence of non-atherosclerotic intracranial stenosis; 7. Previous interventional or surgical treatment in the intra- or extra-carotid artery; 8. Presence of retinal and choroidal diseases such as age macular degeneration (AMD) and moderate to severe cataracts and glaucoma. Demographic and clinical information was recorded for all CAS patients.

### 2.1. Computed Tomography Perfusion Imaging and Post-Processing

CTP imaging was performed on all patients to assess the cerebral blood flow perfusion. A 128-row dual-source CT scanner (Siemens SOMATOM Definition Flash, Siemens Healthcare, Forcheim, Germany) was used for CTP imaging and was initiated 5 s after a contrast agent bolus (350 mg/mL Omnipaque followed by a saline flush of 45 mL at 5 mL/s); Jog mode, 80 kVp/200 mAs; 30 cycles for 45 s and 128 slices. Standard brain reconstruction was performed, and the gantry angle was parallel to and above the orbital roof to avoid radiation exposure to the lens.

### 2.2. Postprocessing of CTP Imaging

CTP images were transferred to a workstation (IntelliSpace Portal system, Philips Healthcare, Amsterdam, The Netherlands) to generate perfusion parameter maps of the cerebral blood flow (CBF), cerebral blood volume (CBV), time to peak (TTP), mean transit time (MTT) and permeability surface (PS). Regions of interest (ROI) were drawn on CTP source images (average images calculated from all phases that could offer accurate anatomical references) and transferred to corresponding parametric maps. Absolute values of CBF, CBV, TTP, MTT and PS in bilateral MCA territories were measured at the basal ganglia level by drawing symmetrical ROI in the two hemispheres on perfusion maps. Relative (r) CTP parameters (rCBF, rCBV, rTTP, rMTT and rPS) were calculated as the ratio of the value on the contralateral side to the ipsilateral side for each ROI as shown in Figure 1. 

### 2.3. SS-OCT/SS-OCTA Imaging

The SS-OCT/SS-OCTA (SVision Imaging, Henan, China) tool had a swept-source laser with a central wavelength of 1050 nm and a scan rate of 200,000 A-scans per second. The tool was equipped with an eye-tracking utility to eliminate eye-motion artifacts. The axial resolution, lateral resolution and scan depth were 5 µm, 13 µm and 3 mm as previously reported [17].

Fundus images were obtained with the OCTA to evaluate the optic nerve head and fundus. Patients with optic nerve head swelling, exudates and other ophthalmic disorders that could affect our data were excluded. Structural OCT imaging of the macula was conducted with 18 radial scan lines centered on the fovea. Each scan line (generated by 2048 A-scans) was 12 mm long and separated by 10°. Automated segmentation of the retinal thickness was conducted with the OCT tool, specifically, the retinal nerve fiber layer (RNFL) and ganglion cell-inner plexiform layer (GCIPL) in a 6 mm^2^ area around the fovea (Figure 2). The mean thicknesses of the retinal structure (measured in µm) were obtained with the OCT tool. The choroidal vascular volume (CVV) was defined as the volume from the basal border of the retinal pigment epithelium-Bruch membrane complex to the choroidoscleral junction (Figure 2).

OCTA images were obtained with a raster scan protocol of 512 horizontal B-scans that covered an area of 6 mm around the fovea. The three-dimensional choroidal vascular index (CVI) was defined as the ratio of the choroidal vascular luminal volume to the total choroidal volume (Figure 2). 

Measurement of visual acuity was conducted with the Snellen chart using light at 2.5 m. For participants with poor vision, finger counting or hand movements were performed and then transformed as visual acuity. The visual acuity of all participants was then transformed to the logarithm of the minimum angle of resolution (LogMAR) for data analyses.

### 2.4. Statistical Analysis

Continuous variables with a normal distribution were expressed as the mean ± standard deviation (SD), while categorical variables were displayed as frequencies and percentages. The z scores of OCT/OCTA were calculated by subtracting the mean value from the value of the observation and dividing by the standard deviation. The comparison of the OCT/OCTA data between controls and CAS patients was conducted with generalized estimating equations (GEE) while adjusting for hypertension, diabetes, dyslipidemia, age, gender and inter-eye dependencies (i.e., left and right eye). A linear mixed model was used to compare the SS-OCT/SS-OCTA parameters between the ipsilateral and contralateral eyes while adjusting for age, gender, hypertension, diabetes and dyslipidemia and inter-eye dependencies. Multiple linear regression with GEE was used to assess the association between SS-OCT/SS-OCTA parameters and CTP parameters while adjusting for risk factors in CAS patients and inter-eye dependencies. A value of *p* < 0.05 was considered statistically significant; SPSS (version 24) was used for all statistical analyses. 

## 3. Results

Fifty-five CAS patients met our inclusion criteria; however, 5 patients were excluded due to the presence of cerebral hemorrhage on MR imaging and poor CTP imaging. Of the 50 patients who underwent SS-OCT/SS-OCTA imaging, 13 were excluded (7 due to age-related macular degeneration, 2 due to severe cataracts and 4 due to poor signal quality). Our data analysis included 37 CAS patients (mean age: 63.95 ± 11.05 years); out of the 37 patients, 32 (86.48%) were males; 22 (59.45%) had hypertension; and 9 (24.32%) had diabetes. Table 1 shows the demographics, ophthalmic and CT perfusion parameters of our CAS patients. Appendix A shows the demographic information and OCT/OCTA parameters between CAS patients and controls. Compared to controls, CAS patients showed thinner retinal thicknesses and reduced choroid (*p* < 0.05, Appendix A). 

### 3.1. Comparison of SS-OCT/SS-OCTA Parameters between Ipsilateral and Contralateral Eyes

Compared with contralateral eyes, ipsilateral eyes showed significantly thinner RNFL (*p* < 0.001, Table 2), GCIPL (*p* = 0.013, Table 2) and CVV (*p* = 0.001, Table 2). Appendix A shows a stenosed and non-stenosed image of the choroid and retina of a CAS patient. 

### 3.2. Correlation between SS-OCTA Parameters and CTP Parameters

rCBV showed a significant correlation with RNFL (*p* < 0.001), GCIPL (*p* < 0.001) and CVI (*p* = 0.027) as shown in Table 3. RNFL (*p* < 0.001) and GCIPL (*p* < 0.001) showed a significant correlation with rPS as shown in Table 3. CVV showed a significant correlation with rCBF (*p* = 0.009), rMTT (*p* < 0.001) and rTTP (*p* < 0.001).

## 4. Discussion

Stenosis in the carotid artery can cause reduced blood flow in the ophthalmic artery which may result in ocular impediment [18,19]. The presence of a microvasculature in the ophthalmic artery makes ocular involvement an uncommon occurrence. Our current study aimed to assess the retinal structural and choroidal changes in CAS patients and their association with cerebral hemodynamic changes. Our current study showed that ipsilateral eyes of patients with carotid stenosis showed significantly thinner RNFL, GCIPL and CVV compared with contralateral eyes. Importantly, our report showed a significant correlation between the retinal structure and choroid and CTP parameters suggesting that imaging the retina and choroid via the OCT/OCTA tool may be a promising tool to reflect the cerebral hemodynamic changes in CAS patients.

A decrease in ocular blood flow in the ophthalmic artery may affect the retinal and choroidal structure. To date, retinal structural reports have been inconsistent. Some reports [20,21] showed that mean RNFL thickness and the ganglion cell complex (GCC) did not differ between CAS patients and controls; contrarily, other reports [22,23] showed RNFL thinning in asymptomatic CAS patients compared with controls. We showed that CAS patients had reduced RNFL and GCIPL thicknesses and reduced CVI and CVV compared to controls which is in line with some previous reports; we also compared the ipsilateral eyes to the contralateral eyes in CAS patients and showed thinner RNFL and GCIPL thicknesses in the ipsilateral eyes compared to contralateral eyes. CAS results in ocular blood flow, which has a significant impact on the structure of the retina. RNFL and GCIPL make up the superficial vascular complex (SVC), which has been suggested to be sensitive to ocular changes in the retina. The significant thinning of the RNFL and GCIPL in the ipsilateral eyes may be due to the reduced blood flow in the ipsilateral eyes as shown in previous retinal microvasculature reports [24,25]. On the other hand, RNFL and GCIPL reflect the neuro-axonal integrity of the retina; we suggest that thinning of the RNFL and GCIPL may indicate neurodegeneration in ipsilateral eyes.

Sayin et al. [21] showed significantly decreased choroidal thickness in CAS, while Rabina et al. showed no significant difference in choroidal thickness of CAS patients compared to controls. In our study, we showed reduced CVV in ipsilateral eyes compared to contralateral eyes which is in line with our previous report [26]. Reduced choroidal vascular volume in the ipsilateral eyes may be linked with hypoperfusion due to irregularities in the choroidal blood flow; this may influence the functionality of the microvascular network which may induce a reduced blood flow pressure, ultimately resulting in the choroidal thinning [27]. Contrarily, reduced CVV in the ipsilateral eyes may be the result of the decreased metabolic activities associated with thinning of the retinal ganglion complex, RGC (which consists of the RNFL and GCL). Likewise, irregularities in the choroidal circulation may cause ischemia in the optic nerve head and retinal microvasculature which may explain the changes that occurred in the ipsilateral eyes of CAS patients. More importantly, the CVV, which reflects the choroidal thickness, is an indicator of ocular health [28,29]; thus, the significant thinning in the ipsilateral eyes may reflect the deterioration of the ocular structures in the ipsilateral eyes.

Choroidal circulation accounts for more than 85% of the ocular circulation [30]; CVI reflects the volumetric vascular density in the choroid [31]. To date, very little is known about the impact of carotid stenosis on choroidal circulation. Previous OCTA reports [16,26] did not find any significant difference in the choriocapillaris (the capillaries of the choroid) before and after stenting in CAS patients; the authors suggested changes seen in the choriocapillaris may be irrelevant in the choroid resulting in the insignificant differences. Similarly, our current report did not find a significant difference in the CVI of ipsilateral eyes compared to contralateral eyes.

Interestingly, we showed that CTP parameters significantly correlated with SS-OCT parameters; rPS and rCBV significantly correlated with RNFL and GCIPL thicknesses in CAS patients. Permeability surface (PS) is an indirect measure of blood-brain barrier (BBB) permeability, while cerebral blood volume (CBV) describes the blood volume of the cerebral capillaries and venules per cerebral tissue volume. The RNFL and GCIPL are in the inner retina which forms the inner blood-retina barrier (iBRB) [32]; importantly, these layers form the superficial vascular complex (SVC) which contains arterioles and venules and is the main blood flow channel into the retina [33]. Changes in the brain are suggested to be mirrored in the retina [34]; thus, the significant correlation between rPS, rCBV, and retinal structural thickness in CAS suggests that changes in the brain may reflect changes in the retina or vice versa.

Changes in the choroid are suggested to reflect changes in the choroidal plexus of the brain [35,36]; both structures contain a microvasculature and serve as a barrier against harmful substances. To the best of our knowledge, this is the first study to examine the association between choroidal changes and cerebral hemodynamic changes in CAS patients. We showed that CVV significantly correlated with rCBF, rMTT and rTTP. CAS is reported to affect the anterior choroidal artery which may affect the choroidal plexus [37] of the brain due to ischemia [38]. Since these CTP parameters reflect the cerebral blood flow, the correlation between the CVV and these CTP parameters may suggest that the choroidal changes may reflect the cerebral blood flow changes in CAS patients. Future studies are needed to validate our speculations.

We would like to acknowledge some limitations in our study. Firstly, our study did not include a comparison group, which should be considered in future studies. The observational study design is another limitation; longitudinal studies are needed to validate our results. Furthermore, our current study assessed the degree of stenosis in all patients; however, other high-risk plaque features such as plaque length, echogenicity and/or surface area may affect the retinal structure and need to be assessed in future studies. Detailed ophthalmic examinations such as optic nerve function tests and the visual field test were not conducted in our study.

In conclusion, we showed that ipsilateral eyes in CAS patients had significantly thinner RNFL, GCIPL and CVV compared with contralateral eyes. We also showed that retinal and choroidal thicknesses correlated with hemodynamic changes assessed with CTP in our CAS group. Taken together, our report suggests that retinal and choroidal changes have the potential to detect hemodynamic changes in CAS patients and could predict the risk of stroke. 

## Figures and Tables

**Figure 1 brainsci-12-00979-f001:**
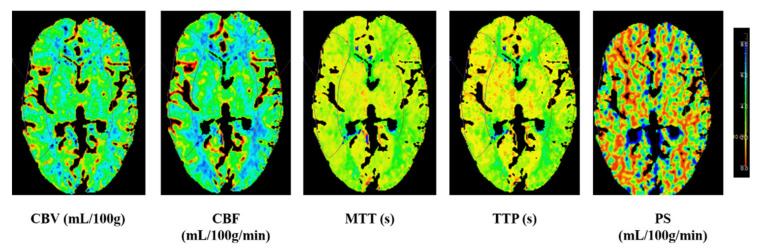
Illustrative image at the basal ganglia level by drawing symmetrical region of interest (ROI) in the two hemispheres on perfusion maps. CBV: cerebral blood volume; CBF: cerebral blood flow; MTT: mean transit time; TTP: time to peak; PS: permeability surface.

**Figure 2 brainsci-12-00979-f002:**
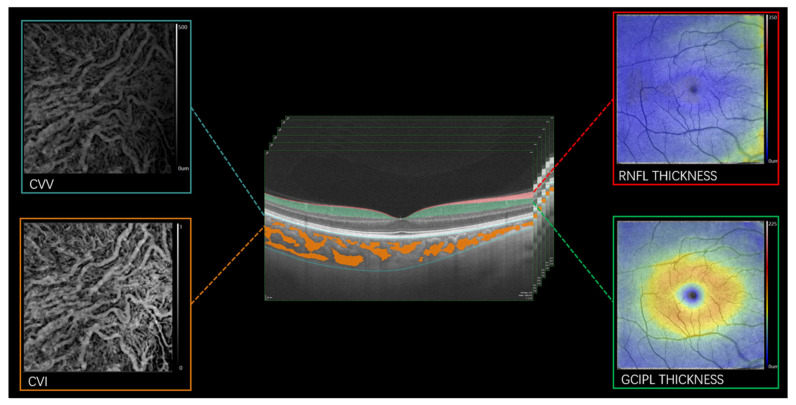
Segmentation of the retinal structure and choroid using the SS-OCT/SS-OCTA. RNFL: retinal nerve fiber layer; GCIPL: ganglion cell-inner plexiform layer; CVI: choroidal vascular index; CVV: choroidal vascular volume.

**Table 1 brainsci-12-00979-t001:** Demographics.

	*n* = 37
Age, years	63.95 ± 11.05
Gender, males	32
Location of stenosis	
Right	23
Left	14
Systolic blood pressure, mmHg	133.76 ± 15.47
Diastolic blood pressure, mmHg	83.68 ± 11.35
Hypertension, *n*	22
Diabetes, *n*	9
Dyslipidemia, *n*	6
Coronary heart disease, *n*	3
Smokers, *n*	24
Drinkers, *n*	18
Ophthalmic parameters	
RNFL, µm	30.46 ± 3.28
GCIPL, µm	68.58 ± 5.23
CVI	0.29 ± 0.06
CVV	0.23 ± 0.09
VA, logMAR	0.17 ± 0.20
Z-RNFL, µm	−0.006 (−0.613–0.774)
Z-GCIPL, µm	0.037 (−0.812–0.907)
Z-CVI	−0.005 (−0.558–0.636)
Z-CVV	−0.047 (−0.527–0.672)
CT perfusion parameters	
rCBV	1.0 (0.94–1.03)
rCBF	1.11 (0.97–1.31)
rMTT	0.88 (0.75–1.01)
rTTP	0.97 (0.92–1.0)
rPS	1.01 (0.90–1.05)

RNFL: retinal nerve fiber layer; GCIPL: ganglion cell-inner plexiform layer; CVI: choroidal vascular index; CVV: choroidal vascular volume; LogMAR: logarithm of minimum angle resolution; r: relative; CBV: cerebral blood volume; CBF: cerebral blood flow; MTT: mean transit time; TTP: time to peak; PS: permeability surface.

**Table 2 brainsci-12-00979-t002:** Comparison of the macula structure and choroidal parameters between ipsilateral and contralateral eyes.

	Ipsilateral	Contralateral	*p*-Value
RNFL, µm	30.19 ± 3.43	30.72 ± 3.16	<0.001
GCIPL, µm	68.14 ± 5.02	68.99 ± 5.45	0.013
CVI	0.29 ± 0.06	0.29 ± 0.06	0.969
CVV	0.23 ± 0.10	0.24 ± 0.10	0.001
VA, logMAR ^Ψ^	0.22 ± 0.23	0.13 ± 0.15	0.062

*p*-values were adjusted for age, gender and vascular risk factors (hypertension, diabetes and dyslipidemia). RNFL: retinal nerve fiber layer; GCIPL: ganglion cell-inner plexiform layer; CVI: choroidal vascular index; CVV: choroidal vascular volume; LogMAR: logarithm of minimum angle resolution; ^Ψ^ ANOVA.

**Table 3 brainsci-12-00979-t003:** Correlation between OCT/OCTA parameters and cerebral hemodynamic parameters.

	RNFL, µm			GCIPL, µm			CVI			CVV		
	B	SE	*p*	B	SE	*p*	B	SE	*p*	B	SE	*p*
rCBV	24.58	5.80	<0.001	41.83	8.95	<0.001	0.11	0.50	0.027	0.14	0.07	0.061
rCBF	6.45	4.51	0.150	4.76	7.31	0.516	0.001	0.03	0.966	0.113	0.04	0.009
rMTT	−2.87	7.03	0.683	−1.31	10.04	0.896	0.021	0.03	0.500	−0.17	0.03	<0.001
rTTP	−7.80	31.19	0.802	31.233	46.46	0.501	0.133	0.18	0.467	−0.61	0.20	0.002
rPS	14.51	3.18	<0.001	31.08	3.25	<0.001	0.046	0.04	0.196	0.064	0.05	0.159

Data were adjusted for age, gender and vascular risk factors (hypertension, diabetes and dyslipidemia). RNFL: retinal nerve fiber layer; GCIPL: ganglion cell-inner plexiform layer; CVI: choroidal vascular index; CVV: choroidal vascular volume; CBV: cerebral blood volume; CBF: cerebral blood flow; MTT: mean transit time; TTP: time to peak; PS: permeability surface; B: beta coefficient; SE: standard error.

## Data Availability

The data that support the findings of this study are available on request from the corresponding author.

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
