# Peer review of "Retinal Thickness Correlates with Cerebral Hemodynamic Changes in Patients with Carotid Artery Stenosis"

_brainsci, 2022, doi:10.3390/brainsci12080979_

Round 1
Reviewer 1 Report
It is an interesting concept but a rather simplistic approach to a very complex question.There is no mention of the ophthalmic evaluation of these eyes.Kindly provide the clinical evaluation details including the visual acuity,optic nerve function tests and fundus findings of these eyes.The cause of decreased RNFL thickness should be explained on the basis of ophthalmic and brain findings and not on the basis of flow abnormalities.
Author Response
It is an interesting concept but a rather simplistic approach to a very complex question.There is no mention of the ophthalmic evaluation of these eyes.Kindly provide the clinical evaluation details including the visual acuity,optic nerve function tests and fundus findings of these eyes.The cause of decreased RNFL thickness should be explained on the basis of ophthalmic and brain findings and not on the basis of flow abnormalities.
Reply: We have added the ophthalmic evaluation we did in our revised manunscript.
Fundus images were obtained with the OCTA to evaluate the optic nerve head and fundus. Patients with optic nerve head swelling, exudates and other ophthalmic disorders that could affect our data were excluded.
Measurement of visual acuity was done with the Snellen chart using light at 2.5 meters. For participants with poor vision, finger counting or hand movements were performed and then transformed as visual acuity. The visual acuity of all participants was then transformed to the logarithm of the minimum angle of resolution (LogMAR) for data analyses.
We also revised the thinning of the RNFL and GCIPL based on ophthalmic findings.
RNFL and GCIPL make up the superficial vascular complex (SVC), which has been suggested to be sensitive to ocular changes in the retina. The significant thinning of the RNFL and GCIPL in the ipsilateral eyes may be due to the reduced blood flow in the ipsilateral eyes as shown in previous retinal microvasculature reports[24, 25]. On the other hand, RNFL and GCIPL reflect the neuro-axonal integrity of the retina; we suggest that thinning of the RNFL and GCIPL may indicate neurodegeneration in ipsilateral eyes.
Optic nerve function tests were not done in our study and was included in our limitation section
Detailed ophthalmic examination such as optic nerve function tests and visual field test were not done in our study.
Reviewer 2 Report
In this study William Robert Kwapong et al. aimed to assess the retinal structural and choroidal changes in carotid artery stenosis patients and their association with cerebral hemodynamic changes. This is an interesting study even if the authors did not include a comparison group. In order to improve the quality of the study I strongly recommend to add the comparison group. Secondly, to better understand the conclusion and to support them I would suggest to add some images related to the retinal and choroidal thicknesses in different scenario mentioneted in the text and in a healthy subject as a comparison.
- page 6 line 209: "strutures in the ipsilateral eyes" please correct with struCtures
Author Response
In this study William Robert Kwapong et al. aimed to assess the retinal structural and choroidal changes in carotid artery stenosis patients and their association with cerebral hemodynamic changes. This is an interesting study even if the authors did not include a comparison group. In order to improve the quality of the study I strongly recommend to add the comparison group. Secondly, to better understand the conclusion and to support them I would suggest to add some images related to the retinal and choroidal thicknesses in different scenario mentioneted in the text and in a healthy subject as a comparison.
- page 6 line 209: "strutures in the ipsilateral eyes" please correct with struCtures
Reply: We would like to that the reviewer for these constructive suggestions. We added a comparison group (ie controls) to our revised data and there was significant changes as seen in our Supplementary Table 1.
With the images, changes between the CAS and controls or changes between the ipsilateral or contralateral eyes cannot be clearly seen on images but can be shown in their data, thus we did not add the images since it cannot help with our data.
Reviewer 3 Report
Dear Authors,
I wish to submit my review of the article titled: "Retinal thickness correlates with cerebral hemodynamic changes in patients with carotid artery stenosis."
The article is well-written and interesting; the authors should be commended for their work.
However, some points should be discussed:
1. In the article, you assessed the different ocular findings comparing both eyes simultaneously (eye on the stenosed carotid artery vs eye on the non-stenosed carotid artery). Measurements obtained from a subject's right and left eye are often correlated, whereas many statistical tests assume observations in a sample are independent. Hence, data collected from both eyes cannot be combined without considering this correlation. Could you please explain how you analyzed the two-eye data and whether you took it into account during the analysis?
2. Statistical Analysis: The article lacks the sample size calculation and the test used for comparison. Are the data normally distributed?
3. Using OCTA could be interesting also to analyze SCP and DCP changes. If available, you should add these data to evaluate the association between retinal vascular changes and cerebral hemodynamic changes.
4. OCTA and OCT images of the eye on the stenosed carotid artery and the eye on the non-stenosed carotid artery should be added.
Author Response
- Comparison of the OCT/OCTA data between controls and CAS patients was done with generalized estimating equations (GEE) while adjusting for hypertension, diabetes, dyslipidemia, age, gender and inter-eye dependencies (i.e left and right eye). A linear mixed model was used to compare the SS-OCT/SS-OCTA parameters between the ipsilateral and contralateral eyes while adjusting for age, gender, hypertension, diabetes, and dyslipidemia and inter-eye dependencies. Multiple linear regression with GEE was used to assess the association between SS-OCT/SS-OCTA parameters and CTP parameters while adjusting for risk factors in CAS patients and intereye dependencies. Inter-eye dependencies was used to adjust for the eye laterality.
2. Thank you for bringing this to our attention. Yes our data was not normally distributed thus we converted our OCT/OCTA data into Z-score as shown in Table 1. The z scores of OCT/OCTA were calculated by subtracting the mean value from the value of the observation and dividing by the standard deviation.
3. Our current study did not analyze the SCP and DCP because recent reports have spoken on it a lot. However, we would keep enrolling more patients and take it up as our next project. Thanks
4. OCT and OCTA images of the stenosed and non-stenosed eye have been added in our Supplementary Image 2. Changes could not be clearly seen in the retinal structure but changes could be seen in the choroidal vessels.
Round 2
Reviewer 1 Report
Any correlation of the OCT features to the CAS is valid ,subject to a normal ophthalmic evaluation in these eyes.It can not be stated in the limitations and we cannot proceed with the analysis unless the ocular examination for the evaluated eyes has been taken in to account.
Very sorry that this paper does not have a scientifically correct methodology.
Reviewer 2 Report
I think now the manuscript could be accepted for publication.
Reviewer 3 Report
Dear Authors,
Thank you for reviewing the article. Your answers fulfill the previous requests.